# Impact of Single-Lumen Versus Double-Lumen Endotracheal Tube on Postoperative Swallowing Function in Lung Transplantation Patients: A Single-Center, Retrospective Cohort Study

**DOI:** 10.3390/jcm14093075

**Published:** 2025-04-29

**Authors:** Rishi Ashok Patel, Haley Nitchie, Bethany J. Wolf, Cecilia Taylor, Loren Francis

**Affiliations:** 1Department of Anesthesiology and Perioperative Medicine, Medical University of South Carolina, Charleston, SC 29425, USA; nitchie@musc.edu (H.N.);; 2Department of Public Health Sciences, Medical University of South Carolina, Charleston, SC 29425, USA

**Keywords:** lung transplant, double-lumen tube, postoperative dysphagia, FOIS score

## Abstract

**Background/Objectives**: The role of double-lumen endotracheal tube (DLT) versus single-lumen endotracheal tube (SLT) use during lung transplantation (LTx) and its effects on postoperative dysphagia have not yet been studied. It has been shown that new-onset oropharyngeal dysphagia (OPD) is common after various thoracic surgeries including lung transplantation and that OPD is associated with increased postoperative complications. **Methods**: A single-center, retrospective cohort study was performed using a data exploration tool in the electronic medical record. Data included demographic characteristics, medical history, postoperative dysphagia measured by Functional Oral Intake Scale (FOIS) via modified barium swallow study (MBSS) within 5 days of surgery, and other secondary outcomes. **Results:** In univariate analysis, participants who had a DLT (49 patients) had significantly higher FOIS scores (indicating better swallowing function) as compared to those with an SLT (21 patients) (*p* = 0.035). Lumen type remained significant in a multivariable model, with use of a DLT showing more than a 5-fold increase in the odds of a higher FOIS score after controlling for other factors (*p* = 0.004; cumulative OR (95% CI): 5.2 (1.7–15.9)). Participants who had a DLT had shorter hospital length of stay (LOS) (*p* = 0.017; single 18 days (IQR = 13), double 14 days (IQR 7)). Those who had a DLT experienced significantly greater ventilator-free time at postoperative day 30 compared to those who received an SLT (*p* = 0.018). ICU LOS was similar between those who received a DLT vs. SLT. **Conclusions**: Overall, DLT seems to confer reduced new-onset OPD after lung transplantation surgery when compared with SLT. The use of DLT instead of SLT for lung isolation for LTx may have the potential to reduce morbidity and mortality in this population.

## 1. Introduction

Lung transplantation (LTx) has dramatically increased in number over the past 20 years with improvements in surgical techniques, immunosuppression, and donor organ management. Several studies have looked at the effects of aspiration in this population on acute and chronic rejection as well as morbidity and mortality [1,2,3]. Bronchiolitis obliterans syndrome (BOS) is the most common form of chronic rejection in LTx, with up to half of recipients developing BOS within five years of transplant [4]. Studies have shown improved outcomes with medical and surgical treatment of gastroesophageal reflux disease [5]. Ultimately, it is the microaspiration of gastric contents that damages airway epithelium and lowers post-transplant allograft survival [6].

New-onset oropharyngeal dysphagia (OPD) is common after various thoracic surgeries including LTx and is associated with increased postoperative complications [7,8]. While there have been extensive discussions of the high incidence of postoperative swallowing dysfunction in lung transplant patients [9] and contributing factors have been identified, direct comparative studies specifically evaluating the effect of DLT versus SLT during LTx surgery on swallowing outcomes are not prominent. OPD will be evaluated using the Functional Oral Intake Score (FOIS) via a modified barium swallow study (MBSS), which has been well-validated in surgical patients [10,11].

## 2. Materials and Methods

This retrospective cohort study was reviewed and approved by the institutional review board (IRB) (Pro00119197; 7 March 2022). Patients who had undergone LTx between 1 January 2016 and 20 April 2022 at the Medical University of South Carolina were identified using a data exploration tool function in the electronic medical record (EMR) system with the following search criteria: (1) admission to cardiac intensive care unit (ICU); (2) lung transplantation surgery. Exclusion criteria included previous history of dysphagia, previous or current tracheostomy, grade 3 or higher primary graft dysfunction, severe preoperative musculoskeletal weakness, and patients that were inpatient at the time they were placed on the transplant list and when the transplant took place.

A standardized form was used to manually extract data from the EMR for those meeting inclusion criteria. Data collection included demographic characteristics (age, gender, weight, body mass index), medical history, postoperative dysphagia measured by FOIS via MBSS within 5 days of surgery, procedure name, operative time (minutes), total ventilation time (minutes), intensive care unit (ICU) length of stay (LOS), hospital LOS, 30-day ventilator-free days, in-hospital mortality, second FOIS score measurement within 10 days of surgery, postoperative FiO_2_ and PaO_2_, tracheostomy, enteral feeding duration (days), and destination on discharge. Following LTx, patients were referred for evaluation of swallowing function by the speech-language pathology team via an instrumental swallowing evaluation and MBSS. Observational findings on instrumental swallowing exams and MBSS were tracked and recorded in the EMR. Patients who did not formally undergo an MBSS were given an international dysphagia diet standardization initiative (IDDSI) score by a speech-language pathologist or other care team member, who evaluated their swallowing function post-operatively within the appropriate time frame. The IDDSI score was then converted to the FOIS score. The IDDSI score assesses the functional diet status of individuals, while the FOIS is an observer-rated scale to document changes in functional oral intake. Studies show a strong correlation between the IDDSI and FOIS. The FOIS score was calculated from the IDDSI score based on the scale from Crary et al. [10,12].

### 2.1. Outcomes

The primary outcome is postoperative dysphagia measured by the initial FOIS via MBSS within 5 days of surgery. Secondary outcomes of interest include ICU LOS, hospital LOS, 30-day ventilator-free days, discharge destination, postoperative FiO_2_ and PO_2_, and use of feeding tube for enteral nutrition.

A priori power: We anticipated a minimum of 60 patients of whom ~25% received SLT and 75% received DLT. A sample size of 15 SLT patients and 45 DLT patients provides 80% power to detect moderate effect sizes of 0.9 using a 2-sided test and significance level α = 0.05. A study conducted by Ito et al. (2017) of thoracotomy patients observed a mean FOIS of 4.9 ± 2.2 in patients 3 months post-surgery [13]. Assuming similar variability in FOIS score is observed in our patients, an effect size of 0.9 corresponds to a 2-point difference in FOIS scores between the two groups.

### 2.2. Statistical Analysis

Descriptive statistics were calculated for all participant and procedural characteristics. Differences in characteristics between patients who had a single vs. a double-lumen tube were examined using 2-sample *t*-tests or Wilcoxon rank sum test for continuous characteristics and chi-square or Fisher’s exact test for categorical characteristics. The primary outcome of interest was the FOIS score at postoperative day 5. Differences in FOIS score by type of endotracheal tube used were initially evaluated using the Wilcoxon rank sum test. To further evaluate the association between tube type and FOIS score, exploratory analyses accounting for additional patient characteristics that might impact FOIS score were conducted using a proportional odds modeling approach. Given the limited number of subjects in the study, participants were grouped into 3 categories of FOIS score for the multivariable proportional odds model: (1) FOIS between 1 and 3 (no oral intake, tube dependent with minimal oral intake, and tube supplements with consistent oral intake), FIOS of 4 or 5 (total oral intake of a single consistency, total oral intake of multiple consistencies requiring special preparation), and FIOS of 6 or 7 (total oral intake with no special preparation but must avoid specific foods or liquid items, total oral intake with no restriction). Interactions between covariates with type of tube used were also examined in the multivariable model. The proportional odds assumption was examined to ensure model assumptions were met. All covariates univariately associated at *p* < 0.2 with the 3-scale FOIS score defined above were considered in the model. Given the exploratory nature of the proportional odds model, all covariates with *p* < 0.1 were retained in the multivariable model.

Secondary outcomes of interest include discharge location, ICU LOS, hospital LOS, and number of ventilator-free days at postoperative day 30. Differences in discharge location by endotracheal tube type were evaluated using Fisher’s exact test and differences in all other outcomes were evaluated using the Wilcoxon rank sum test. All analyses were conducted in SAS v 9.4 (SAS Institute, Cary, NC, USA).

## 3. Results

The study included 70 patients, of whom 49 had a double-lumen tube and 21 had a single-lumen tube. A majority of participants were male (60%) and the mean age of participants was 57.5 ± 12.3 years. Relative to participants who had received a single-lumen tube intraoperatively, those who had a double-lumen tube were younger (*p* = 0.006), less likely to have cardiopulmonary bypass (*p* = 0.050), and had shorter intraoperative cardiopulmonary mechanical support duration (*p* = 0.002). This, however, did not translate to a difference in operative time or total ventilation times for each group.

The groups were similar in terms of proportion of males, BMI, use of intra- and postoperative extracorporeal membrane oxygenation (ECMO), occurrence of tracheostomy, operative time, and ventilation time. Participant and procedural characteristics by intraoperative lumen tube type are shown in Table 1. All patients had similar surgical techniques as the operations were performed by two surgeons working in cohort, this is further supported in Table 1 with groups having similar total operative times. Intraoperative management was also similar between the groups; all patients received general anesthesia with standard monitors and utilized a transesophageal echocardiogram probe for monitoring. The patients all had the same extubation protocols postoperatively and were continually assessed for a spontaneous breathing trial and trial of extubation. The total ventilation time did not differ significantly between groups.

*FOIS score*: In univariate analysis, participants who had a double-lumen tube had significantly higher FOIS scores (indicating better swallowing function) compared to those with a single-lumen tube (*p* = 0.035). The median FOIS score on POD 5 in those with a double-lumen tube was 7 (IQR = 2) and in those with a single-lumen tube was 5 (IQR = 4). The distribution of FOIS score on POD 5 by type of lumen tube used is shown in Figure 1.

We also examined the association between tube type and FOIS score on a 3-point scale accounting for additional patient characteristics using a proportional odds modeling approach. In a univariate proportional odds model, FOIS score was associated with type of lumen tube used. Specifically, use of a double-lumen tube was associated with a 2.9-fold increase in the odds of observing a higher level of FOIS (better swallowing function) compared to single-lumen tubes (*p* = 0.035; cOR: 2.90, 95% CI: 1.08, 7.82). Those patients who had intraoperative ECMO had lower odds of having a higher FOIS score (*p* = 0.057; cOR: 0.31, 95% CI: 0.06, 1.04). Longer operative time and longer duration of cardiopulmonary mechanical support also showed odds of a lower FOIS score (operative time: *p* = 0.144; cOR for 1 h increase: 0.74; 95% CI: 0.50, 1.11; CPB time: *p* = 0.100; cOR for 10 min increase: 0.94, 95% CI: 0.88, 1.01), though neither was significant. Participant age, sex, BMI, operative time, and ventilation time were not associated with FOIS in univariate models.

Variables considered in the multivariable cumulative logit model included type of lumen tube, operative time, cardiopulmonary bypass time, intraoperative ECMO use, and the interactions between lumen tube type with all other variables. Variables retained in the final model at significance *p* < 0.1 included lumen tube type, total operative time, and use of intraoperative ECMO. Use of a double-lumen tube was associated with more than a 5-fold increase in the odds of a higher FOIS score after controlling for intraoperative ECMO and operative time (*p* = 0.004; cOR: 5.2; 95% CI: 1.68, 15.9). Longer total OR times showed decreased odds of a higher FOIS score adjusting for lumen tube type and ECMO, though statistical significance at *p* < 0.05 was not achieved (*p* = 0.086; cOR for 1 h increase: 0.69; 95% CI: 0.45, 1.05). Intraoperative ECMO use was associated with FOIS score such that those who received intraoperative ECMO had 0.18 times the odds of a higher FOIS score compared to those who did not, controlling for lumen tube type and total operative time (*p* = 0.013; cOR: 0.18, 95% CI: 0.05, 0.70). These results are shown in Table 2.

*Secondary Outcomes*: We also examined associations between discharge location, number of ventilator-free days at postoperative day 30, ICU length of stay, hospital length of stay, postoperative FiO_2_ and PO_2_, and enteral feeding. Participants who had a double-lumen tube had shorter hospital length of stay (*p* = 0.017; single 18 days (IQR = 13), double 14 days (IQR 7)). Those who had a double-lumen tube had a significantly greater amount of ventilator-free time at postoperative day 30 compared to those who received a single-lumen tube (*p* = 0.018). Discharge location and ICU length of stay were similar between those who received a double vs. a single-lumen tube. Patients with double-lumen tubes had lower FiO_2_ postoperatively, higher PO_2_/FiO_2_ ratio postoperatively (*p* = 0.038), and were less likely to require enteral feeding (*p* = 0.050), as shown in Table 3. The distribution of hospital LOS and number of ventilator-free days at postoperative day 30 by lumen tube type are shown in Figure 2 and Figure 3.

## 4. Discussion

There are no studies to date that evaluate postoperative dysphagia in patients undergoing lung transplantation when comparing double- and single-lumen endotracheal tubes. Lung transplantation surgery requires lung isolation to facilitate recipient pneumonectomy followed by implantation of either one or two donor lungs. Cardiopulmonary bypass (CPB) or extracorporeal membranous oxygenation support may be used, and mechanical support choice may impact lung isolation strategy. Research suggests a correlation between endotracheal tube size and the risk of post-extubation dysphagia. One study in acute respiratory failure survivors found that larger endotracheal tube sizes (≥8.0 mm) were associated with an increased risk of aspiration and the development of laryngeal granulation tissue, though this study was not specific to lung transplant patients. Additionally, intubation with large or double-lumen endotracheal tubes can cause injury to the larynx [14]. DLTs have larger external diameters than single-lumen tubes; consequently, we hypothesized that this size difference could influence postoperative dysphagia outcomes. DLTs allow for continuous positive airway pressure application to the operative lung and better suctioning of that lung than a smaller diameter bronchial blocker through an SLT. A DLT also allows for easy conversion from one- to two-lung ventilation and is less prone to moving out of position than a bronchial blocker in an SLT. Given these advantages, DLT is often preferred over SLT for procedures requiring lung isolation. Potential disadvantages of DLT may be related to their comparatively large size and rigidity, which may make them more difficult to place in a patient with a challenging airway or cause complications including local tissue trauma, sore throat, and vocal cord injuries [12].

SLTs can also be utilized for lung isolation with the addition of a bronchial blocker. Advantages of utilizing SLT with bronchial blockers include ease of placement in patients with difficult airway anatomies such as obesity, limited neck extension, or distorted anatomy, as well as the pediatric patient population [15]. The presence of an SLT prevents the need to re-secure the airway at the end of the case. Disadvantages include a multistep process for insertion which leads to a longer positioning time when compared to DLT and the possible need for repositioning if switching which lung is isolated [15].

Lung isolation strategy is one area where anesthesiologists may be able to modify risk factors for lung transplantation patients [16]. With postoperative dysphagia in mind, it was noted in this study that participants who had a DLT had significantly higher FOIS scores (corresponding with better swallowing function) compared to those with an SLT (*p* = 0.035). Use of a DLT was associated with a 2.9-fold increase in the odds of observing a higher level of FOIS compared to SLT (*p* = 0.035). After controlling for intraoperative ECMO and operative time, this increased to more than a 5-fold increase in the odds ratio of observing a higher FOIS score (*p* = 0.004). Also, participants who received a double-lumen tube had shorter hospital length of stay (*p* = 0.017; single 18 days (IQR = 13), double 14 days (IQR 7)) and a significantly greater amount of ventilator-free time at postoperative day 30 compared to those who received a single-lumen tube (*p* = 0.018).

These results are somewhat surprising as the literature has shown an association between endotracheal tube size and aspiration, with larger (>8.0) ETT being associated with increased incidence of aspiration, albeit utilizing single-lumen endotracheal tubes [14]. One explanation for our findings could be that the size of the double-lumen ETT was carefully selected for the specific patient, ensuring proper fit, versus placing 8.0 single-lumen tubes regardless of patient size in order to accommodate bronchoscopy and placement of a bronchial blocker. The medical literature has many studies to determine the correct DLT size for patients but there is no consensus. Duthie et al. reported that in 96.3% of their patients, the correct DLT size could be predicted using the height and gender of the patient [17]. Standard textbooks state for women shorter and taller than 160 cm, 35 Fr and 37 Fr DLT should be used, respectively [18]. Similarly for men shorter and taller than 170 cm, 39 Fr and 41 Fr DLT should be used [18]. Another explanation could be that DLT provided more efficient and consistent lung isolation and better surgical exposure; however, this did not translate into a difference in operative time between groups. Taken together, the use of DLT over SLT showed significantly improved swallow score, more ventilator-free time, reduced hospital length of stay, and lower rates of postoperative feeding tube use.

Our study has several weaknesses including its single-center, retrospective, and nonrandomized design, which potentially limits the external validity of our findings. Our findings may not be generalizable to all centers due to differences in surgical protocols, patient populations, or perioperative protocols. Our sample size is overall small (70), with the majority of patients having DLT (49). Given the imbalance between groups, our study is subject to chance bias which could limit generalizability. While improvement in swallow score was statistically significant, this result could have been skewed by that distribution. The majority of patients were male. Males, on average, have larger airways, which can allow for better ease of placement of DLT. However, this effect may have been mitigated given that various sizes of DLT are available. Also, the DLT group overall had younger patients which may have skewed the results.

Given our results above, there is ample opportunity for prospective studies in the future to explore the difference between lung isolation techniques in this population. A prospective, randomized, parallel-group control trial comparing dysphagia outcomes between DLT and SLT postoperatively could further aid in reducing postoperative complications in this population. Anesthesiologists tailor the lung isolation strategy (e.g., double-lumen tube vs. bronchial blocker) for each lung transplant patient based on anatomical considerations, airway examination, and surgical planning. Our results, which characterize the postoperative dysphagia risk associated with specific devices, could potentially inform this selection, perhaps leading clinicians to more readily consider DLTs within their risk/benefit assessment for dysphagia.

## 5. Conclusions

Overall, DLT seems to confer reduced new-onset dysphagia after lung transplantation surgery when compared with SLT. DLTs are also associated with improved 30-day ventilator-free days and hospital length of stay in this population. While anesthesiologists select lung isolation methods for lung transplant patients based on established factors like anatomy, airway, and surgical plan, awareness of the postoperative dysphagia risks presented herein might shift the consideration toward DLT when evaluating the patient’s overall risk profile.

## Figures and Tables

**Figure 1 jcm-14-03075-f001:**
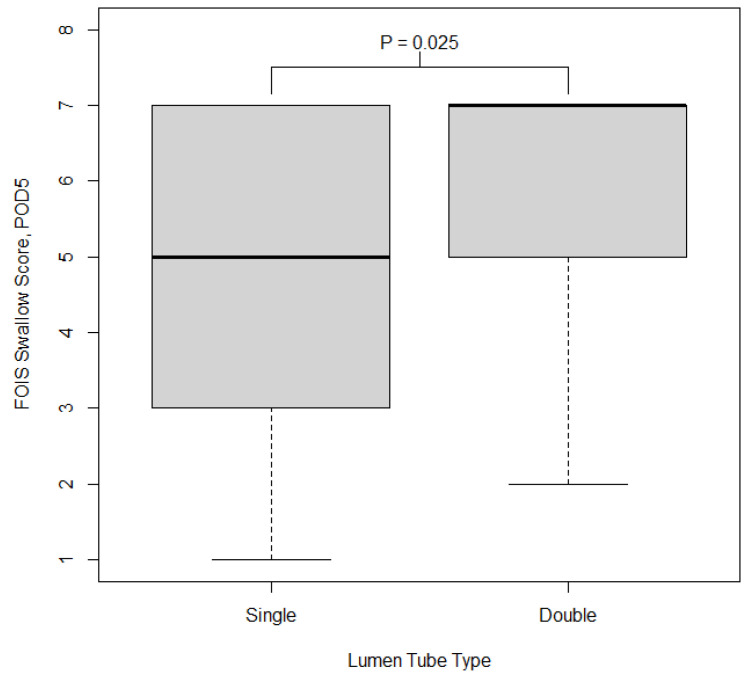
Boxplot of FOIS score on postoperative day 5 by type of lumen tube used. Boxes show the 25th, 50th, and 75th percentiles in each group; whiskers extend 1.5 times the inner quartile range from the median. The *p*-value reported on the plot is from the Wilcoxon rank sum test comparing FOIS between groups.

**Figure 2 jcm-14-03075-f002:**
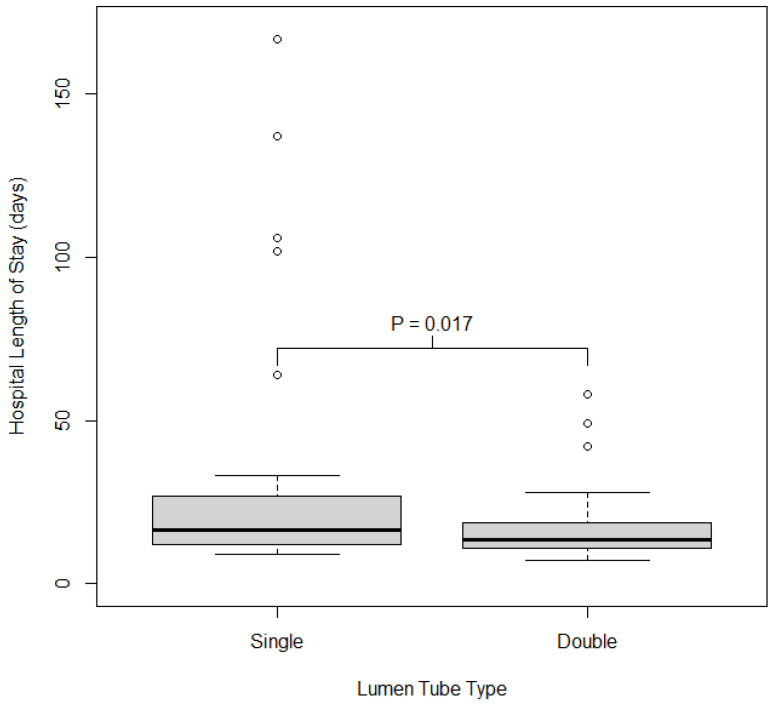
Distribution of hospital length of stay in days by type of lumen tube used. Boxes show the 25th, 50th, and 75th percentiles in each group; whiskers extend 1.5 times the inner quartile range from the median. The *p*-value reported on the plot is from the Wilcoxon rank sum test comparing HLOS between groups.

**Figure 3 jcm-14-03075-f003:**
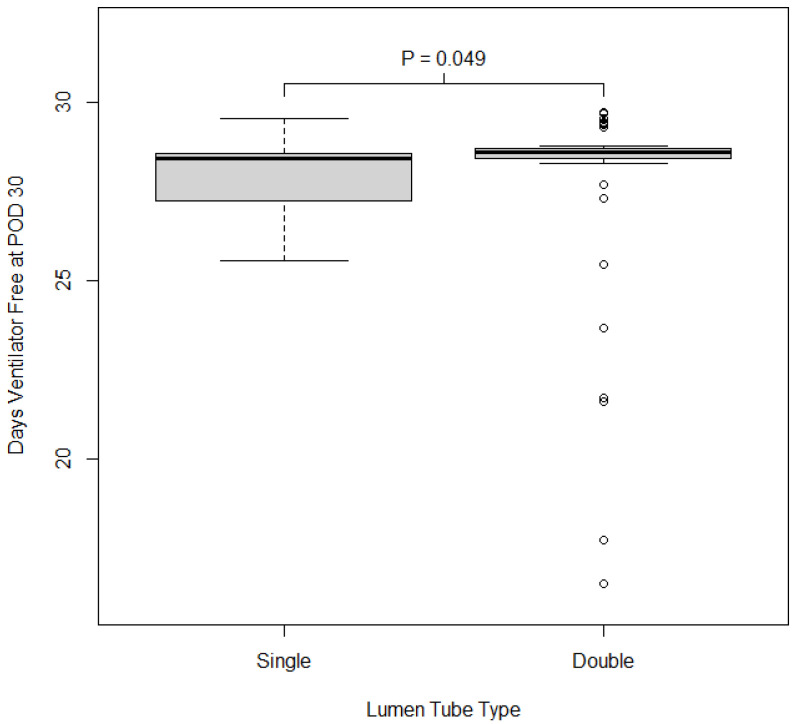
Distribution of days ventilator-free at postoperative day 30 type of lumen tube used. Boxes show the 25th, 50th, and 75th percentiles in each group; whiskers extend 1.5 times the inner quartile range from the median. The *p*-value reported on the plot is from the Wilcoxon rank sum test comparing FOIS between groups.

**Table 1 jcm-14-03075-t001:** Patient and procedural characteristics by lumen tube type. Categorical variables are reported as n (%) and continuous variables are reported as mean (SD) * or median (IQR; min, max) ^†^ as appropriate.

Characteristic	Single (n = 21)	Double (n = 49)	*p*
Age, years *	63.6 (8.10)	54.9 (12.9)	0.006
Sex, Male	14 (66.7)	28 (57.1)	0.456
BMI, kg/m^2^ *	26.8 (3.39)	25.3 (3.88)	0.119
Lung transplant type			0.212
Left lung	2 (9.52)	1 (2.04)	
Both	19 (90.5)	48 (98.0)	
Any ECMO, Yes	2 (9.52)	14 (28.6)	0.121
Intraoperative ECMO, Yes	1 (4.76)	10 (20.4)	0.154
Postoperative ECMO, Yes	1 (4.76)	4 (8.16)	1.000
Cardiopulmonary bypass, Yes	20 (95.2)	36 (73.5)	0.050
Intraoperative cardiopulmonary mechanical support time, min ^†^	163 (45; 87, 456)	124 (81; 0, 294)	0.002
Tracheostomy, Yes	3 (14.3)	3 (6.12)	0.355
Operative time, min ^†^	378 (89; 237, 499)	377 (93; 104, 626)	0.380
Ventilation time, min ^†^	1800 (2306; 592, 49,928)	1195 (1086; 400, 85,420)	0.183

**Table 2 jcm-14-03075-t002:** Cumulative odds ratios for the univariate and multivariable cumulative logit models.

	Univariate		Multivariable	
Predictor	cOR (95% CI)	*p*	cOR (95% CI)	*p*
Type of Lumen Tube, Double vs. Single	2.90 (1.08, 7.82)	0.035	5.17 (1.68, 15.9)	0.004
Operative time, 1 h increase	0.74 (0.50, 1.11)	0.144	0.69 (0.45, 1.05)	0.086
Intraop ECMO, Yes vs. No	0.31 (0.09, 1.04)	0.057	0.18 (0.05, 0.70)	0.013
Intraop CPB, Yes vs. No	2.38 (0.78, 7.24)	0.126		
CPB time, 10 min increase	0.94 (0.88, 1.01)	0.100		
Age, 1 year increase	1.01 (0.97, 1.05)	0.718		
Sex, Male vs. Female	0.87 (0.34, 2.25)	0.780		
BMI, 1 kg/m^2^ increase	0.98 (0.87, 1.11)	0.779		
Time on Ventilator, 1 h increase	0.999 (0.997, 1.001)	0.328		

**Table 3 jcm-14-03075-t003:** Secondary outcomes by type of lumen tube.

Outcome	Single (n = 21)	Double (n = 49)	*p*
Discharge Destination, n (%)			1.000
Home	14 (66.7)	32 (65.3)	
Rehabilitation Center	2 (9.52)	4 (8.16)	
Other	5 (23.8)	13 (26.5)	
30-Day ventilator-free days, median (IQR; range)	28.4 (1.1; 0, 29.5)	28.6 (0.29; 0, 39.3)	0.049
ICU length of stay, days, median (IQR; range)	6 (5; 3, 140)	6 (3; 3, 48)	0.643
Hospital length of stay, days, median (IQR; range)	18 (13; 10, 167)	14 (7; 7, 102)	0.017
FiO_2_ 72 h Postop	40 (10; 20, 60)	25 (11; 21, 100)	0.005
PO_2_ arterial 72 h Postop	84 (57; 36, 227)	86 (27; 38, 376)	0.796
PO_2_/FiO_2_ ratio	255 (187.6; 71.7, 905)	314.3 (87.5; 140, 850)	0.038
Any Enteral Nutrition via Feeding Tube, Yes	16 (76.2)	25 (51.0)	0.050
If Yes, duration in days	8.5 (10; 1, 146)	5 (8; 1, 82)	0.532
If Yes, # enteral feedings	7.5 (8; 1, 31)	5 (9; 1, 28)	0.873

## Data Availability

The data can be provided upon reasonable request.

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
