# Peer review of "Impact of Single-Lumen Versus Double-Lumen Endotracheal Tube on Postoperative Swallowing Function in Lung Transplantation Patients: A Single-Center, Retrospective Cohort Study"

_jcm, 2025, doi:10.3390/jcm14093075_

Round 1
Reviewer 1 Report
Comments and Suggestions for Authors
Manuscript Number: jcm-3554032
Title: Impact of Single Lumen versus Double Lumen Endotracheal Tube on Postoperative Swallowing Function in Lung Transplantation Patients
Thank you very much for giving me the opportunity to review your paper. I have read the manuscript entitled “Impact of Single Lumen versus Double Lumen Endotracheal Tube on Postoperative Swallowing Function in Lung Transplantation Patients." which provides information on the effects of double lumen endotracheal tube on postoperative swallowing function in lung transplantation patients.
The paper presents a relevant and timely research topic, making a valuable contribution to the field. The research objective is clearly defined, and the study follows a systematic methodology, ensuring reproducibility. The data sources are reliable, and the analysis methods are well-explained. Additionally, the logical flow of the paper makes it easy to follow from introduction to conclusion.
However, there are several areas that require improvement.
Title
Include the type of research in the title
Introduction
If the study does not clearly differentiate itself from previous research, it may be considered redundant. Reviewers might question whether the topic has already been sufficiently explored.
The introduction could better justify the necessity of the study by emphasizing how it differs from previous research. Clearly outlining the study’s uniqueness would enhance its impact.
Methods
A lack of detailed methodological descriptions can reduce the study’s reproducibility and credibility. Without clear explanations, reviewers may doubt the reliability of the findings.
The experimental methods section should provide more details to ensure reproducibility. Including specific information on the equipment used, data collection procedures, and experimental conditions would make the methodology more transparent.
Results
If the data sources are unclear or the sample size is too small, the validity of the study may be questioned. Weak data reliability can lead to doubts about the accuracy of the conclusions.
Simply presenting results without a thorough discussion of their significance weakens the impact of the study. Reviewers may find the analysis incomplete or lacking depth.
The results section would benefit from a more in-depth discussion. Instead of simply presenting the findings, the paper should interpret their statistical significance and broader implications. Using visual aids such as graphs and charts would further improve clarity.
Conclusion
The conclusion should better highlight the study’s contributions and its broader implications. Emphasizing how this research advances knowledge in the field would make the paper more impactful.
The language and expression in some parts of the paper could be refined for better readability. Revising unclear or awkward sentences and ensuring concise yet precise explanations would enhance the overall quality of the manuscript.
Additionally, the references should be updated to include more recent studies, preferably within the last five years. Incorporating the latest research would strengthen the credibility of the study.
Overall, the study is well-structured and provides meaningful insights, but refining the introduction, methodology, results, language, references, and conclusion would significantly improve its quality. Let me know if you need more specific feedback on certain sections.
Author Response
Dear Reviewer,
Thank you very much for taking the time to review this manuscript. Please find the detailed responses below in blue and the corresponding revisions/corrections highlighted in the re-submitted files.
Reviewer #1
Comment 1: Include the type of research in the title
Response 1: We have adjusted the title of the article. [Page 1, Line 4]
Comment 2: The introduction could better justify the necessity of the study by emphasizing how it differs from previous research. Clearly outlining the study’s uniqueness would enhance its impact.
Response 2: Thank you for pointing this out. We have adjusted the introduction to better justify the necessity of the study. [Page 2, Paragraph 3, Lines 47-50]
Comment 3: The experimental methods section should provide more details to ensure reproducibility. Including specific information on the equipment used, data collection procedures, and experimental conditions would make the methodology more transparent.
Response 3: Thank you for your comment. We would like more specific feedback on details needed to ensure reproducibility. In our methods section, we outlined the study’s retrospective nature, use of data exploration tool in the electronic medical record system to extract patient information, and the data collected manually through a standardized form. We have also updated the methods section to outline the calculation of the FOIS score. [Page 3, Paragraph 2, Lines 81-84]
Comment 4: The results section would benefit from a more in-depth discussion. Instead of simply presenting the findings, the paper should interpret their statistical significance and broader implications. Using visual aids such as graphs and charts would further improve clarity.
Response 4: Thank you for this comment. We discuss the findings of the study and interpret their statistical significance in the discussion section in the paper. We have also included Tables 1-3 and Figures 1-3 as visual aids. We would appreciate more specific feedback on our analysis.
Comment 5: The conclusion should better highlight the study’s contributions and its broader implications. Emphasizing how this research advances knowledge in the field would make the paper more impactful.
Response 5: Thank you for your comment. We agree and so have adjusted our conclusions to better highlight the study’s broader implications and impact in the field of lung transplantation. [Page 10, Paragraph 4, Lines 297-300]
Reviewer 2 Report
Comments and Suggestions for Authors
This study addresses a clinically relevant and underexplored question: the impact of endotracheal tube type (SLT vs DLT) on postoperative dysphagia following lung transplantation. The hypothesis that DLT may be associated with better swallowing function is both interesting and clinically plausible. Although retrospective, the study is well-structured and utilizes validated assessment tools such as FOIS and MBSS. However, several limitations and areas for improvement are identified and discussed below.
-Strengths of the Study:
1. Well-defined clinical question with potential implications for perioperative anesthetic practice and the multidisciplinary management of lung transplant recipients.
2. Use of validated, objective outcome measures (FOIS and MBSS) for dysphagia assessment.
3. Appropriate exclusion criteria, enhancing sample homogeneity.
4. Clinically relevant outcomes, including not only swallowing function but also secondary variables such as duration of mechanical ventilation, ventilator-free days, and hospital length of stay.
*Methodological Limitations and Suggestions:
1. Single-center retrospective design: this limits the external validity of the findings. Although acknowledged in the discussion, it should be more clearly emphasized as a primary limitation.
2. Unequal group distribution (SLT vs DLT): while sample size justification and univariate analyses are presented, there may be selection bias. A multivariate analysis adjusting for potential confounders (age, comorbidities, surgical approach, ventilation duration, etc.) would strengthen the conclusions.
3. Conversion from IDDSI to FOIS: the methodology for this conversion is insufficiently detailed. A table outlining equivalencies and supporting references should be included to minimize variability and increase transparency.
4. Lack of surgical and anesthetic protocol details: it is unclear whether intraoperative management, surgical technique, or extubation protocols differed between groups. These factors could significantly influence postoperative swallowing outcomes and should be addressed.
5. Formatting errors in the manuscript (e.g., “Error! Reference source not found.”): these must be corrected before publication to meet editorial standards.
*Conclusion:
This is a well-conceived study with findings relevant to clinical practice in the context of lung transplantation. Despite the inherent limitations of its retrospective design, the results suggest a potential benefit of double-lumen over single-lumen endotracheal tubes in terms of postoperative swallowing function. Additionally, a more in-depth discussion of the potential pathophysiological mechanisms underlying the observed association is encouraged.
Author Response
Dear Reviewer,
Thank you very much for taking the time to review this manuscript. Please find the detailed responses below in blue and the corresponding revisions/corrections highlighted in the re-submitted files.
Reviewer #2
Comment 1: Single-center retrospective design: this limits the external validity of the findings. Although acknowledged in the discussion, it should be more clearly emphasized as a primary limitation.
Response 1: Thank you for your comment. We have specifically included this limitation in our discussion. [Page 10, Paragraph 2, Line 277-278]
Comment 2: Unequal group distribution (SLT vs DLT): while sample size justification and univariate analyses are presented, there may be selection bias. A multivariate analysis adjusting for potential confounders (age, comorbidities, surgical approach, ventilation duration, etc.) would strengthen the conclusions.
Response 2: We thank the reviewer for this comment, and we agree that a multivariable analysis provides stronger evidence. Notably, we did include a multivariable analysis of the primary outcome for FOIS score in the original submission and this is described in the “Statistical Analysis” section and presented in the results in Table 2. While we do present the univariate results for full range of FOIS scores (1-7), we conducted a multivariable analysis of FOIS score grouped into 3 categories: FOIS between 1 and 3, FIOS of 4 or 5, and FIOS of 6 or 7. We used a cumulative logit model approach to evaluate the modified 3-point FOIS score as it is appropriate for modeling ordinal outcomes while allowing for adjustment of potentially confounding variables. The modified version of FOIS was used in the multivariable analysis because there were a limited number of subjects in each of the 7 levels and this yielded insufficient numbers of subjects with FOIS level to fit a cumulative logit model. The multivariable model did account for operative time and use of intraoperative ECMO which were selected based on association with the outcome and the magnitude of the association between lumen tube type and FOIS score was even stronger after adjusting for these variables.
The abstract in our initial submission did focus on the univariate results and we have edited the abstract to reflect the multivariable analysis. [Page 1, Paragraph 1, Lines 23-26]
Comment 3: Conversion from IDDSI to FOIS: the methodology for this conversion is insufficiently detailed. A table outlining equivalencies and supporting references should be included to minimize variability and increase transparency.
Response 3: Thank you for this comment. We have added to the methods section to detail the conversion from IDDSI to FOIS. There is precedent for this in the literature as the IDDSI assesses the functional diet status of individuals, while the FOIS is an observer-rated scale to document changes in functional oral intake. Studies show strong correlation between IDDSI and FOIS scores; and the FOIS score can be calculated based on the scale from Crary et al. [Page 3, Paragraph 1, Lines 78-81]
Comment 4: Lack of surgical and anesthetic protocol details: it is unclear whether intraoperative management, surgical technique, or extubation protocols differed between groups. These factors could significantly influence postoperative swallowing outcomes and should be addressed.
Response 4: Thank you for this comment. We have updated the results section to specify surgical and anesthetic protocol details. All patients had similar surgical techniques as they were performed by two surgeons working in cohort, this is further supported in Table 1 with groups having similar total operative times. Intraoperative management was also similar in groups, all patients received general anesthesia and utilized a transesophageal echocardiogram probe for monitoring. The patients all had the same extubation protocols postoperatively and were continually assessed for a spontaneous breathing trial and trial of extubation. The total ventilation time did not differ significantly between groups. [Page 4, Paragraph 3, Lines 137-144]
Comment 5: Formatting errors in the manuscript (e.g., “Error! Reference source not found.”): these must be corrected before publication to meet editorial standards.
Response 5: Thank you for pointing this out. We have fixed the reference formatting errors in the introduction. [Page 2, Paragraph 2, Lines 39-44]
Reviewer 3 Report
Comments and Suggestions for Authors
Thank you very much for this interesting manuscript which evaluate postoperative dysphagia in patients undergoing lung transplantation when comparing double and single lumen endotracheal tubes.
I think that the discussion should not start with the sentence "To the authors' knowledge.."; moreover change "author's" into "authors'"
This sentence could be moved before the weakness paragraph.
In my opinion the main limitation of this study is that the double lumen endotracheal tube group had younger patients, about 9 years younger.
However, the double lumen endotracheal tube group had more patients who used extracorporeal membrane oxygenation, although not statistically significant.
Obviouslly, retrospective studies could show differences between the groups studied.
Many thanks
Author Response
Dear Reviewer,
Thank you very much for taking the time to review this manuscript. Please find the detailed responses below in blue and the corresponding revisions/corrections highlighted in the re-submitted files.
Comment 1: I think that the discussion should not start with the sentence "To the authors' knowledge.."; moreover change "author's" into "authors'"
Response 1: We agree and have adjusted the first sentence in our discussion. [Page 9, Paragraph 1, Line 217]
Round 2
Reviewer 1 Report
Comments and Suggestions for Authors
Manuscript Number: jcm-3554032
Title: Impact of Single Lumen versus Double Lumen Endotracheal Tube on Postoperative Swallowing Function in Lung Transplantation Patients
This paper has faithfully supplemented the key points pointed out in the previous version, and the current version is suitable for publication in the journal. However, limitations in sample size and design should be supplemented in future studies, and this may be the reason for a reviewer's minor revision.
Here are the remaining concerns that, while not critical, should still be addressed or acknowledged clearly when submitting the paper for publication:
- Limited Sample Size and Group Imbalance
The total sample size is relatively small (n=70), with a significant imbalance between the DLT group (n=49) and the SLT group (n=21). Although the authors acknowledge this in the discussion, reviewers may still raise concerns about potential statistical bias or reduced generalizability. Consider including a more explicit statement on how this imbalance might affect interpretation and emphasize the exploratory nature of the study.
-Single-Center Design and External Validity
The study was conducted at a single institution with a specific surgical team, limiting the external applicability of the findings. Clarify in the discussion that results may not be fully generalizable to all centers due to differences in surgical protocols, patient populations, or perioperative care.
-Limited Discussion on Future Study Design
While the need for future prospective research is mentioned, the manuscript lacks a more detailed proposal or suggestion for what such studies should specifically address. Briefly outline what an ideal prospective, multicenter study could look like (e.g., randomized design, stratified sampling) to add depth to the conclusion and highlight the importance of follow-up work.
Author Response
Dear Reviewer,
Thank you very much for taking the time to review this manuscript. Please find the detailed responses below in blue and the corresponding revisions/corrections highlighted in the re-submitted files.
Reviewer
Comment 1: The total sample size is relatively small (n=70), with a significant imbalance between the DLT group (n=49) and the SLT group (n=21). Although the authors acknowledge this in the discussion, reviewers may still raise concerns about potential statistical bias or reduced generalizability. Consider including a more explicit statement on how this imbalance might affect interpretation and emphasize the exploratory nature of the study.
Response 1: Thank you for your comment. We have added a more explicit statement in the discussion to further explain this limitation in our study. [Page 9, Paragraph 4, Lines 282-283]
Comment 2: The study was conducted at a single institution with a specific surgical team, limiting the external applicability of the findings. Clarify in the discussion that results may not be fully generalizable to all centers due to differences in surgical protocols, patient populations, or perioperative care.
Response 2: Thank you for your comment. We have adjusted this portion in the discussion to better reflect the external applicability of our findings. [Page 9, Paragraph 4, Lines 279-281]
Comment 3: While the need for future prospective research is mentioned, the manuscript lacks a more detailed proposal or suggestion for what such studies should specifically address. Briefly outline what an ideal prospective, multicenter study could look like (e.g., randomized design, stratified sampling) to add depth to the conclusion and highlight the importance of follow-up work.
Response 3: We agree with your comment. We have expanded our discussion to include a more detailed proposal to highlight the importance of the follow-up work needed. [Page 9, Paragraph 5, Lines 289-292]